# Definition, Epidemiology, Pathophysiology, and Essential Criteria for Diagnosis of Pediatric Chronic Myeloid Leukemia

**DOI:** 10.3390/cancers13040798

**Published:** 2021-02-14

**Authors:** Meinolf Suttorp, Frédéric Millot, Stephanie Sembill, Hélène Deutsch, Markus Metzler

**Affiliations:** 1Pediatric Hemato-Oncology, Medical Faculty, Technical University Dresden, D-01307 Dresden, Germany; 2Inserm CIC 1402, University Hospital Poitiers, F-86000 Poitiers, France; f.millot@chu-poitiers.fr (F.M.); Helene.DEUTSCH@chu-poitiers.fr (H.D.); 3Pediatric Oncology and Hematology, Department of Pediatrics and Adolescent Medicine, University Hospital Erlangen, D-91504 Erlangen, Germany; Stephanie.Sembill@uk-erlangen.de (S.S.); Markus.Metzler@uk-erlangen.de (M.M.)

**Keywords:** pediatric CML, diagnostic essential criteria, epidemiology, pathophysiology

## Abstract

**Simple Summary:**

The low incidence (1:1,000,000) of chronic myeloid leukemia (CML) in the first two decades of life presents an obstacle to accumulation of pediatric experience and knowledge on this leukemia. Biological features of CML are shared but also differing between adult and pediatric patients. This review aims; (i) to define the disease based on an unified terminology, (ii) to list the diseases to be considered as a differential diagnosis in children, (iii) to outlines the morphological, histopathological and immuno-phenotypical findings of pediatric CML, (iv) to illustrate rare but classical complications resulting from high white cell and platelet counts at diagnosis, and (v) to recommend a uniform approach for the diagnostic procedures to be applied. Evidently, only a clear detailed picture of all relevant features can lay the basis for standardized treatment approaches.

**Abstract:**

Depending on the analytical tool applied, the hallmarks of chronic myeloid leukemia (CML) are the Philadelphia Chromosome and the resulting mRNA fusion transcript BCR-ABL1. With an incidence of 1 per 1 million of children this malignancy is very rare in the first 20 years of life. This article aims to; (i) define the disease based on the WHO nomenclature, the appropriate ICD 11 code and to unify the terminology, (ii) delineate features of epidemiology, etiology, and pathophysiology that are shared, but also differing between adult and pediatric patients with CML, (iii) give a short summary on the diseases to be considered as a differential diagnosis of pediatric CML, (iv) to describe the morphological, histopathological and immunophenotypical findings of CML in pediatric patients, (v) illustrate rare but classical complications resulting from rheological problems observed at diagnosis, (vi) list essential and desirable diagnostic criteria, which hopefully in the future will help to unify the attempts when approaching this rare pediatric malignancy.

## 1. Introduction

For a long time, the rarity of chronic myeloid leukemia (CML) in minors has hampered the accumulation of in-depth knowledge from cases presenting in the first two decades of life. Before the introduction of tyrosine kinase inhibitors (TKI) for the treatment of CML, stem cell transplantation was the recommended therapeutic approach for young patients and data on pediatric CML were deposited at the EBMT and IBMTR registries [1,2,3]. Given the enormous improvement of therapeutic success achieved by TKIs, the interest in the long-term outcome of this novel treatment especially in pediatric patients resulted in foundation of the International Registry on Pediatric CML (IR-PCML) at Poitiers/France in the year 2010 [4]. Since then, the number of collaboration centers and in parallel of registered patients continuously has increased. As of today, data on more than 660 patients diagnosed with CML at a median age of 12 years (range 0–17 years) have been collected. The information depicted from a registry on a rare disease like CML in minors offers the enormous benefit to enable treating physicians to apply a uniform approach to diagnose and follow-up this leukemia. Based on the collected data and the shared experience also in very small subcohorts (e.g., CML at very young age, diagnosis in advanced stage of disease, treatment failure) this article recommends a uniform approach for the diagnostic procedures to be applied in the management of CML in children and young adolescents.

## 2. Definition of CML

CML BCR-ABL1 positive is an acquired clonal myeloproliferative hematological malignancy derived from an abnormal pluripotent bone marrow stem cell. The leukemic cell clone consistently is characterized by a specific cytogenetic anomaly the so-called Philadelphia (Ph1) chromosome representing a reciprocal chromosomal translocation t(9;22)(q34.1;q11.2) which generates the BCR-ABL1 fusion gene. Cryptic translocations -being invisible on banding chromosome preparations- or variant translocations involving other chromosomes may represent an obstacle when establishing a diagnosis of CML. The presence of the Ph1 chromosome or BCR-ABL1 sharply separates CML from other myeloproliferative neoplasms (MPNs) like essential thrombocytosis (ET), polycythemia vera (PV) and idiopathic (osteo)myelofibrosis (OMF/IMF) [5]. Notably, the detection of the Ph1 chromosome is not sufficiently specific to diagnose CML, as it is also found in acute lymphoblastic leukemia (2–5% of pediatric cases of ALL).

The BCR-ABL1 is present in the bone marrow in all myeloid lineages as well as in some lymphoid cells [6,7]. Whether endothelial cells of the bone marrow niche are BCR-ABL1 positive is a matter of debate [8,9,10]. Morphologically, CML is characterized by a hypercellular bone marrow, an unregulated growth of myeloid cells (neutrophils, eosinophils, basophils and megakaryocytes) resulting in abnormally high level of morphologically terminally differentiated granulocytes, as well as myeloid precursor cells in the blood and is associated with splenic enlargement in >60% of affected children.

The term “Pediatric” CML does not only indicate the age of a patient affected from CML BCR-ABL1 but also points to distinct biological features (see below Section 9) which are observed with decreasing frequency from the 3rd to 6th decade of life. The latter represents the age when CML BCR-ABL1 is typically diagnosed in Caucasian adults.

## 3. Related Terminology and ICD Codes

Synonyms of CML BCR-ABL1 positive are listed in Table 1. These terms are based on the laboratory method used to establish the diagnosis (based on chromosomal analysis or on molecular technique) and the terms to describe the myeloproliferative characteristics in histological findings. The term “myeloid” should be used in the English literature on pediatric CML for the sake of brevity and replace the terms “granulocytic” or “myelogenous”. The term “Juvenile CML” should not be used at all to avoid any possibility of confusion and mixing-up pediatric CML with the completely different entity of juvenile myelomonocytic leukemia (JMML) [11]. JMML is a unique pediatric disorder also different from chronic myelomonocytic leukemia (CMML) in adults [5,12,13].

The WHO’s system for International Classification of Diseases (ICD) in the present version of ICD11 recommends the codes as listed in Table 2 for categorization of CML. These codes do also apply in children with CML.

## 4. Staging and Classification of CML by Phases

Historically, based on the quantity of blasts, CML is categorized into three progressive phases driving the aggressiveness of the disease.
-Chronic phase (CML-CP) is the most common, indolent clinical stable phase of CML lasting for several years. The myeloid cells are differentiated, with less than 10% of blast cells present in the bone marrow. The response to therapy is excellent.-If untreated, CML-CP usually progresses to the accelerated phase (CML-AP). The cells multiply aggressively and the blast cells increase to 10–19%. Additional chromosomal aberrations beside the Ph+ may be detectable [14]. The response to therapy becomes poorer.-From CML-AP the leukemia progresses to a blastic phase (CML-BP) which is indistinguishable from acute leukemia exhibiting >20% (or ≥30%, see below) of bone marrow blasts of either myeloid or lymphoid immunophenotype. The response to therapy is very poor.

Logically the identification of the phase of CML forms the basis for treatment planning. However, the quantitative morphological criteria as established by the WHO [15,16] and the European LeukemiaNet (ELN) [17] as listed in Table 3 differ for CML-AP and CML-BP. For example, the WHO-recommended criteria for CML-BP are ≥20% of blast cells in blood or bone marrow, extramedullary blast proliferation, or large foci or clusters of blasts in the bone marrow biopsy while the ELN threshold is set to 30% of blasts. In adults, the borderline range has a clinical impact as in a comparative analysis, adult patients who had a blast percentage of 20–29% which is considered CML-BP according to the WHO classification, had a significantly better response rate (21% vs. 8%) and 3-year survival rate (42% vs. 10%) compared with patients who had blasts ≥ 30% [18].

In the TKI era, treatment response may also be used to classify CML-AP (provisional WHO definition) [15] such as:hematologic resistance to the first TKI (or failure to achieve a complete hematologic response to the first TKI) **or**any hematological, cytogenetic, or molecular indications of resistance to 2 sequential TKIs **or**occurrence of 2 or more mutations in BCR-ABL1 during TKI therapy

The ELN criteria have been recommended previously [19] by the international Berlin-Frankfurt-Muenster (BFM) group also for pediatric CML as these criteria have been used in the majority of randomized clinical trials when treating CML in adults with TKI. Guidelines recently published by the Children’s Oncology Group from North America [20] are using the criteria of the National Comprehensive Cancer Network (NCCN) [21] which are derived from the WHO criteria.

In addition, it should be noticed that the introduction of new treatments could change the boundaries between CP, AP, and BC, and modify to some extent the classic subdivision of CML into three phases. Both classification systems agree on that independently from the proportion of blasts any extramedullary infiltration of organs beside liver and spleen in CML must be classified as blastic phase.

## 5. Subtypes of CML

One large breakpoint region (approx. 200 kb) is found in the ABL1 gene on chromosome 9q34 whereas three breakpoint regions are present in the so-called breakpoint cluster region (BCR) gene on chromosome 22q11 (Figure 1). Like in adults, in pediatric CML the vast majority of breakpoints cluster in a small region of the major (M-bcr) breakpoint while alternate, less common breakpoints cluster in the minor bcr (m-bcr) and very rarely further upstream in the BCR gene, in the so-called micro bcr (µ-bcr) [22,23,24]. M-bcr, m-bcr, and µ-bcr are associated with the p190, p210 and p230 BCR–ABL1 fusion proteins, respectively [25]. These three well-defined breakpoint regions in the BCR gene can produce at least eight different m-RNA fusion transcripts (M-bcr, p210: e14a2, e13a2, e14a3, e13a3; m-bcr, p190: e1a2, e1a3; µ-bcr, p230: e19a2, e19a3) because of alternative splicing in the ABL1 gene (splicing to exon 2 or exon 3) and because the M-bcr consists of two intronic regions (intron 13 and intron 14) [25,26]. 

In CML-CP, the transcript types e13a2 and e14a2 are present with a frequency of 95%. In a single center study on pediatric CML (N = 146 patients), a proportion of 38% patients harbored transcript type e13a2 and 36% transcript e14a2 while the remaining 26% patients expressed both transcripts due to alternate splicing [22]. In a worldwide analysis on patients of all ages (N = 45,503 patients) transcript e13a2 is detected more frequently in males (39.2%) than in females (36.2%) and correlates with age, decreasing from 39.6% in children and adolescents down to 31.6% in patients ≥ 80 years old [27]. Whether or not differences resulting from the presence of either transcript e14a2 or e13a2 exert a possible impact on the treatment response is discussed below in the Section 16. In addition, several very rare BCR–ABL1 variant fusion genes (resulting in the p195, p200 and p225 BCR–ABL1 fusion proteins; fusion transcripts e6a2, e8a2, and e18a2, respectively) have been detected -partly in single cases [26,27,28,29,30].

## 6. Differential Diagnosis

If CML is suspected but the Ph1 chromosome and BCR-ABL1 fusion transcript are absent, non-malignant disorders with a clinical and hematological picture mimicking CML need to be excluded first. ***Leukemoid reaction*** (LR) is the major differential diagnosis of CML in patients presenting with a leukocyte count in the range of 50,000 cells/µL and significant increase in mature neutrophils with a marked shift to the left [31]. Laboratory findings like toxic granulocytic vacuolation, Döhle’s bodies in the granulocytes, absence of basophilia, and a normal or increased leukocyte alkaline phosphatase (LAP) score separate the LR from CML [32]. Basophils are normal in LR and splenomegaly is an unusual finding. Taking carefully the clinical history and physical examination is suggestive of the origin of the LR, which is very heterogeneous comprising infections (especially *S. aureus*, *S. pneumoniae*), inflammatory syndromes (e.g., glomerulonephritis), malignancies, drugs (corticosteroids can cause a short-lasting extreme left-shifted neutrophilia), intoxications (liver failure), severe hemorrhage, and acute hemolysis [33].

Without karyotyping, CML may be more difficult to differentiate from the other classical myeloproliferative neoplasms (MPNs) occurring in adults which are comparatively uncommon in children. ***Polycythemia vera*** (PV) with associated iron deficiency (e.g., teenage girls with hypermenorrhagia), which causes normal hemoglobin and hematocrit values, can manifest with leukocytosis and thrombocytosis. Isolated megakaryocytic hyperplasia can be seen in ***Essential Thrombocythemia*** (ET) with marked thrombocytosis and splenomegaly. Such patients usually have a normal or increased LAP score, a WBC count less than 25,000/µL, and no Ph1 abnormality. However, concerning the typical mutations a lower incidence of mutation JAK2-V617F has been reported in childhood ET and PV, and fewer CALR mutations were found in children with ET [34]. ***Primary (osteo)myelofibrosis*** (PMF) also termed idiopathic myelofibrosis (IMF) is extremely rare in children [35], although sporadic childhood cases with PMF/IMF have been described [36]. Compared to adults, phenotypic differences appear to exist in children with PMF/IMF which are typically also found in pediatric CML, such as a frequent presence of marrow eosinophilia, only a low degree of marrow collagen fibrosis, the absence of significant osteosclerosis and megakaryocytic dysplasia with hypolobulated megakaryocytes with hyperchromatic nuclei and micromegakaryocytes. Notably, in none of 40 cases pediatric cases with IMF reported worldwide a previously described mutation, such as JAK2-V617F has been identified [37].

Virtually all cases of pediatric ***Myelodysplastic Syndrome*** (MDS) present with pancytopenia involving all three cell lineages, while single lineage cytopenia or macrocytosis may occasionally be the presenting findings [5]. Contrasting CML, leukocytosis is generally not a feature of MDS. Some patients may present with moderate hepatosplenomegaly but most have no organomegaly. Cytogenetic aberrations (monosomy 7, trisomy 8, 5q-, trisomy 21) are found in 55%–75% children with MDS. In childhood, the bone marrow may be hypo- or normocellular but rarely hypercellular. While small megakaryocytes are also found in CML, typical dysplastic features like macrocytic erythropoiesis, unusually large megakaryocytes, and dysgranulopoiesis point towards MDS but are not found in CML.

***Juvenile Myelomonocytic Leukemia*** (JMML) typically manifests in infants and thereafter until the fifth year of life with declining incidence. Patients present with fever, infection, pallor, bleeding, hepato- or splenomegaly, lymphadenopathy, and skin rash [5]. The blood smear is characteristic showing uniformly elevated WBC count with absolute mo-nocytosis, anemia, and thrombocytopenia and is often more helpful in diagnosing than the BM morphology in which monocytosis is often only discrete. An additional finding not detected in CML is an increased HbF in patients with normal karyotype [11]. Mutations are found in the Ras signal transduction pathway downstream of the receptor in about 90% of patients; thus, JMML belongs to the group of diseases called RASopathies (Noonan syndrome, CBL-syndrome, NF1) [38].

Extremely rarely, pediatric patients may present with myeloid hyperplasia, which involves almost exclusively the neutrophil, eosinophil, or basophil cell lineage. These patients are described as having ***chronic neutrophilic, eosinophilic, or basophilic leukemia*** and do not have evidence of the Ph1 chromosome or the BCR-ABL1 gene. The World Health Organization defines MPD with eosinophilia and constitutively activated platelet-derived growth factor receptor-α, or -β, or fibroblast growth factor receptor 1 as a distinct category [39]. Occasionally, these myeloid neoplasms present in the first years of life with leukocytosis and organomegaly, and thus need to be differentiated from JMML or CML with increased eosinophils [11].

## 7. Epidemiology

CML usually presents at a median age of 60 years in Caucasians, but at younger age (35–45 years) in Asians [40,41,42]. It is rare in children contributing only 2–3% of all pediatric leukemia cases. The global incidence rate of CML is 15/1,000,000 per year with a male to female ratio of 1.34 while the age-adjusted incidence rate for the age group <18 years is 1.0 per 1,000,000 [43]. In the first three years of life pediatric CML is extremely rare [44]. The SEERS database showing pooled data on all myeloproliferative diseases at childhood age lists a continuously increasing incidence rate from 0.7 cases in the age group 1 to 4 years up to 4.3 cases per 1,000,000 at 15 to 19 years (Table 4). From the authors’ experience, myeloproliferative diseases besides CML like essential thrombocytosis, polycythemia vera and myelofibrosis are more than 10-fold less frequent than CML in the first two decades of life. Therefore, the SEERS data give a rather detailed impression on the continuous increase in the incidence of CML especially in the 2nd decade of life.

## 8. Etiology

Predisposing factors to pediatric CML are not known. Ionizing radiation is considered a rare risk factor in adults. The maximum increase in CML incidence was observed in atomic bomb survivors in Hiroshima after a median time of 6 years, however, not after the Chernobyl nuclear power plant accident. Probably only exposure to higher radiation doses causes CML. Following irradiation and/or chemotherapy applied in the context of the treatment of a malignancy—mostly Hodgkin and Non-Hodgkin lymphomas—CML has been observed as secondary malignancy rarely in some adult and pediatric cases [46,47,48,49]. As the incidence of pediatric CML is not increased in healthy siblings, and especially not in twin pairs with one child affected from CML, genetic factors are of greater importance in the etiology of CML [50,51,52,53]. The role of mutated so-called myeloid “driver” genes is increasingly getting into the focus in pediatric CML [54,55].

## 9. Pathogenesis

The acquisition of BCR-ABL1 in a hematopoietic stem cell drives its transformation to become a leukemic stem cell (LSC). The fusion protein BCR-ABL1 represents a constitutive active tyrosine kinase considered to be the pathogenic driver capable of initiating and maintaining the disease. BCR-ABL1 activates a number of oncogenic signaling pathways, including PI3K/AKT/mTOR, RAS/RAF/MEK/ERK, and JAK/STAT [56]. However, numerous papers have described that the BCR-ABL1 oncogene does not operate alone when driving disease emergence, maintenance, and progression [57,58]. In children with CML, the myeloid driver mutation ASXL1 is found more frequently than in adult CML [54].

Additional biological differences in adult CML comprise a single breakpoint cluster within the first centromeric 1.5 kb of the BCR gene, whereas in pediatric CML there is a bimodal breakpoint distribution which is similar to adult Ph+ ALL harboring the less frequently observed M-BCR rearrangement [59]. In pediatric CML the bimodal breakpoint distribution in the BCR gene changes to the adult pattern at the age of 13 years, probably in association with the onset of puberty (Figure 2). For so far unknown reasons, the adult type breakpoint distribution pattern at prepubertal age is found more frequently in girls, but not in boys. We hypothesize that there are uncharacterized sex differences in the non-coding regions of the genome on which changes in the sex hormone blood levels starting at puberty exert an influence.

In pediatric acute leukemias the origin of the malignant cell clone has been traced back to its development in utero by using dried blood spots archived at birth on Guthrie cards [60,61]. The first case of pediatric CML in which BCR-ABL1, but no blood abnormalities were present at birth was identified recently from an archived cord blood specimen retrospectively when the infant presented with CML-CP at the age of 6 months [62]. A high-sensitive technique using nested genomic DNA-based PCR with subject-specific PCR primers after characterizing the genomic breakpoints [59,63,64] identified a clonal burden of less than 1 in 10,000 leukocytes at birth. In this case, a mutation associated with pre-disposition to several myeloid cancers was ruled out based on a defined myeloid panel of 11 driver genes (ACD, ANKRD26, CEBPA, DDX41, ETV6, GATA2, RUNX1, SRP72, TERC, TERT, and TP53). If all cases of CML –like acute leukemias– would also take their origin already in utero the observed sex difference at puberty concerning the BCR breakpoint could hardly be explained.

BCR-ABL1 can also be detected –rarely but with age-dependent increase from newborn to older age– in blood specimen collected from healthy individuals [65]. Thus, the dogma of BCR-ABL1 representing the sole event initiating CML is challenged [66]. In all patients receiving long-term TKI-treatment, CML-LSCs persist as they are resistant to the effects of TKIs. Bone marrow microenvironment-generated signals, cell autonomous BCR-ABL1 kinase-independent genetic changes, and epigenetic alterations all contribute to: (i) persistence of a quiescent LSC reservoir, (ii) innate or acquired resistance to TKIs, and (iii) progression into the fatal blast crisis stage [67,68].

## 10. Clinical Features and Hematological Findings

There are no organ alterations protruding as signs specific for CML. The bone marrow and blood are generally involved in pediatric CML. Besides hypersplenism, CML may cause fever, infection, easy bleeding, mild normocytic anemia, fatigue, bone pain, or other unspecific symptoms [44,69,70]. Compared to adults, pediatric CML-CP presents with more aggressive clinical and biological features such as a higher proportion of patients exhibiting splenomegaly, a larger spleen size, and higher leukocyte and platelets counts [69,71,72,73].

More than 90% of all pediatric patients with CML are diagnosed in CML-CP. The proportion of children in advanced phases (CML-AP and CML-BP) represents only 7.5% of all patients according to the International Registry for Childhood CML [4,74]. Because of leukemic infiltrates the spleen is enlarged in 70% to 80% and the liver in 50% to 60% of pediatric patients in CML-CP [69,70,75]. In CML-BP, however, any tissue (lymph nodes, skin, soft tissue, bones, and CNS) may be infiltrated by blasts [76]. Solid extramedullary manifestations of CML historically are termed chloroma because of the greenish color caused by the presence of myeloperoxidase [77]. Significantly, any extramedullary organ infiltration (except of spleen, liver, or retinal infiltration) results in upstaging a patient from CML-CP or CML-AP to CML-BP. CNS infiltration is not seen in CML-CP but the diagnosis of CML-BP requires i.th. prophylactic treatment [19,78].

CML is diagnosed in one third of pediatric patients incidentally when a blood count is performed to clarify other medical conditions [70,79]. Compared to adult CML, the mean leukocyte count in pediatric CML at diagnosis is more than four-fold higher (60 × 10^9^ cells/L versus 240 × 10^9^ cells/L) [70,71,79,80]. Therefore, once anticoagulated blood from pediatric patients with CML is allowed to separate into cellular elements and plasma, a broad “buffy coat” (“leukocrit”) overtopping the size of the hematocrit is usually visible. Mild normocytic, normochromic anemia (median Hb 10.4 g/dL) is present at diagnosis in 60% pediatric patients in CML-CP [69,70].

The WBC differential usually shows granulopoietic cells at all stages of maturation, from myeloblasts to mature, morphologically normal granulocytes. This “pathological shift to the left” is typical for CML and as outlined in Figure 3 easily allows the separation from other conditions like reactive shift to the left caused by infections or from AML with presence of blasts but missing more mature granulopoietic cells (hiatus leucemicus). Neutrophil function in CML is normal or only mildly impaired. These mature granulocytes have decreased apoptosis, resulting in accumulation of long-lived cells with low or absent enzymatic activity, such as alkaline phosphatase. Basophils are usually elevated in the range of 5% to 10% in the peripheral blood and eosinophils may be mildly increased as well.

Contrasting findings in acute leukemias, the platelet count is normal in CML-CP and even elevated above 500 × 10^9^/L in half of the pediatric patients [81]. Thrombosis results extremely rarely from this alteration -instead mucocutaneous bleeding is observed in more than 10% children with elevated platelet counts. Bleeding is caused by a reduced plasma concentration of large Von Willebrand (VW) factor multimers, indicating a diagnosis of acquired VW-syndrome, which resolves after initiation of CML treatment. Platelet function abnormalities like Glanzmann thrombasthenia in CML at diagnosis have also been described in the literature [82].

During the accelerated phase of CML the proportion of immature cells increases and thrombocytopenia usually develops. Basophils may increase, and granulocyte maturation becomes defective.

## 11. Rheological Problems Associated with High White Cell and Platelet Count

Despite the high white cell count in children at diagnosis, rheological problems are observed in less than 10% of children presenting with an extremely high median WBC of 458,000/µL [83]. Carefully history taking and physical examination is required in children admitted with CML because aside from typical symptoms of leukostasis like dyspnea, headache and dissziness, also blurred vision due to retinal bleeding and infiltrates with leukemic cells (Figure 4) (generally reversible within months) [84] can be observed, as well as low flow/venoocclusive priapism in boys (caveat: emergency scenario to avoid irreversible erectile dysfunction) [85,86,87], or hearing impairment (highly probably irreversible) [88]. Precautions to avoid tumor lysis syndrome include initial administration of fluid (dose 2 L/sqm body surface). NaHCO_3_ may be used to adjust urine pH within a range of pH 6.4 to pH 6.8 to optimize excretion of uric acid and addition of allopurinol in those cases with elevated uric acid serum levels.

Splenomegaly is observed in two thirds of all pediatric patients. Imaging procedures show no grossly visible nodules; however, splenic infarcts may appear if the organ is massively enlarged (Figure 5). Rarely, cases with splenic rupture have been reported in adults, but so far, not in children [89].

## 12. Histopathological Findings

Comparable to CML in adults, the marrow in pediatric CML-CP is significantly hypercellular (95%–100%) presenting with markedly increased granulopoieses and a virtual absence of adipocytes (Figure 6). The ratio myeloid to erythroid cells is ≥10:1 with usually an impressively predominant proportion of myelocytes, promyelocytes and segmented neutrophils; the ratio of myeloblasts in CML-CP by definition may be up to 10%. Mega-karyocytic proliferation is present in >50% of the cases with micromegakaryocytes (hypolobated nuclei) (Figure 6). Typically, these findings are accompanied by immature eosinophils and basophils [91] (Figure 7). Eosinophils with atypical basophilic staining granules known as a “Harlequin” cell are also most commonly seen in CML [92]. Pseudo Gaucher cells and see-blue histiocytes can be found inhomogenously distributed in the marrow aspirate smears (Figure 8) in approximately one third of pediatric patients [93]. Grades of fibrosis are varying and manifest myelofibrosis has been described as an adverse morphological factor in adult CML which may be detected in 15% of pediatric CML [94].

By definition in CML-AP myeloblasts make up 10% –19% of marrow nucleated cells. Often massive basophilia (>20% basophils) is found. An increased number of megakaryocytes is organized in clusters with expressive fibrosis detectable. CML-BP is indistinguishable from acute leukemia representing either myeloblasts (AML) or lymphoblasts (ALL). Sometimes transformation into acute myelomonocytic leukemia or rarely into biphenotypic/bilinear leukemia is diagnosed.

## 13. Cytology/Immunophenotype

The leukocyte (neutrophil) alkaline phosphatase (LAP) level as a marker of terminally differentiated neutrophilic granulocytes is decreased in CML [95,96]. Antigens normally found on neutrophils (CD15, HLA-DR) may be expressed weakly. However, cytochemistry is no longer important in the diagnosis of CML as cytogenetic and molecular genetic analysis permit a much more precise diagnosis. In routine diagnostics, immunophenotyping has no useful role during CML-CP. However, in CML-BP blasts of lymphoid lineage –either upfront at diagnosis of CML or during treatment developing under imatinib—are found more frequently than in adults [97]. Notably, CML-BP-lymphoid is associated with a better prognosis than CML-BP-myeloid [74,78].

CML-BP-lymphoid is usually of B-lineage expressing precursor B lymphoblastic antigens (CD10+, CD19+, CD34+, TdT+, sIg−), but cases of precursor T-cell origin (CD3+, CD7+, TdT+) have been described, too [97,98]. The co-expression of myeloid antigens on lymphoid blasts is frequent (Figure 9). In CML-BP-myeloid myeloperoxidase may be expressed variably (missing, weak, strong), but will show antigens associated with myeloid, monocytic, megakaryocytic, and/or erythroid differentiation. Not uncommonly, one or more lymphoid antigens are also expressed. Rarely, CML-BP blasts of lymphoid and myeloid lineage are present simultaneously [99]. 

## 14. Diagnostic Assessments

Morphology and cytogenetics from a bone marrow aspirate still form the absolutely essential basis of diagnosis [19,20,100,101]. Determining the proportion of blast cells by morphology will assess the correct stage (CML-CP, -AP, -BP), while cytogenetics by complete banding assay of Giemsa-stained metaphases cells identifies the Ph1 chromosome. Karyotyping is also important to rule out additional chromosomal aberrations serving as a warning sign of a poorer prognosis [14,19]. A trephine biopsy should be done in addition to evaluate the degree of fibrosis and may identify nests of blasts, not evident in the aspirate, which both have prognostic impact [94,102,103,104]. If a molecular assay (see below) demonstrates BCR-ABL1, but the Ph1 chromosome cannot be identified by cytogenetics, a FISH test is required.

At diagnosis, performing a reverse transcriptase PCR (RT-PCR) assay on peripheral blood cells is mandatory to identify the type of BCR-ABL1 transcript as a marker for molecular monitoring in response to TKI-treatment [105]. Up to 5% of patients may harbor atypical BCR-ABL1 transcripts lacking ABL1 exon2 or resulting from atypical BCR breakpoints (see Section 5) [27]. Using routine primer/probe sets may yield a false negative PCR in qualitative or quantitative RT-PCR protocols thus hampering the subsequent assessment of molecular response [106,107].

## 15. Essential and Desirable Diagnostic Criteria

The following criteria are essential to diagnose CML BCR-ABL1:-Full blood count with manual differential for percent of blasts, promyelocytes, eosinophils, and basophils. It should be performed before any therapy has been given and is useful to assess the correct stage of the disease and to calculate prognostic risk scores-spleen size, liver size (both in cm below the costal margin) on quiet breathing,-assessment of extramedullary manifestation of CML (lymph nodes, skin, bone, etc.),-demonstration of the Ph+ chromosome, including assessment of additional chromosomal aberrations, loss of chromosomes, derivative chromosome 9,-demonstration of the BCR-ABL1 fusion gene,-type of fusion gene transcript type as detected by RT-PCR

Assessment of the following criteria is desirable when CML BCR-ABL1 is diagnosed:
(a) for optimal monitoring of treatment response in individual patients:-gender, age, height, bodyweight in correlation to the TKI dose administered [108],-identification of mutations in the BCR-ABL1 kinase domain in patients with CML-AP and CML-BP,-identification of the BCR-ABL1 breakpoint on a genomic (DNA) level [63,66,109,110,111,112],(b) to compare data on pediatric CML BCR-ABL1 internationally:-a threphine biopsy (degree of fibrosis, nests of blasts),-due to the rarity of pediatric CML all patients should be enrolled in trials and enrolled into the international pediatric CML registry [4],-for comparison of BCR-ABL1 mRNA levels when assessing the treatment response, data derived from individual laboratories must be aligned to a reference method (International Standard, IS) applying laboratory-specific conversion factors [113,114],(c) to improve the scientific understanding of the disease:-identification of genes and their products influencing TKI blood serum concentration and metabolism [108,115],-assessment of acquired von Willebrand disease in cases with elevated platelets [73,81],-vaccination status at diagnosis and maintenance of immunity under TKI treatment [116,117,118],-identification of somatic or germline mutations and epigenetic modification in addition to BCR-ABL1 [55,57,66,112,119,120,121,122].


## 16. Prognosis and Prediction

The prognosis of adult patients with CML could be predicted with prognostic scores (e.g., Sokal, Hasford, EUTOS) based on the clinical and biological characteristics of the disease at diagnosis [120,123]. The usefulness of these scores has not been formally demonstrated in children. However, the EUTOS Long Term Survival (ELTS) score showed better differentiation in terms of progression free survival than other scores in children with CML and could be incorporated into therapeutic strategy for childhood CML [72]. Moreover, kinetics of decreasing transcript levels within the first months of treatment could identify children requiring an alternative treatment strategy [124].

As can be depicted from Figure 1 (see above, Section 5), due to the additional exon e14, the mRNA chain of transcript e14a2 is longer than of transcript e13a2. The length of the corresponding translated protein p210e14a2 differs by 25 amino acid residues coded by the e14 exon and an amino acid substitution (Glu903Asp) [24]. Several studies analyzed in adult patients with CML whether the two transcripts e14a2 and e13a2 have different responses to imatinib treatment (for an overview see [125,126]). According to the majority—but not all—studies published, patients with e13a2 transcript treated with imatinib have lower and slower cytogenetic and molecular responses than those with an e14a2 transcript. In pediatric cohorts with CML the frequency of the two transcript types has been analyzed showing a slightly higher incidence of e14a2 than in adults. However, the prognostic relevance and significance of either transcript type for therapy or long-term survival have not been clearly defined due to the rarity of the disease [70,127].

## 17. Summary and Conclusions

The rarity of CML in pediatrics results in little experience even in physicians involved for prolonged periods in the treatment of childhood cancers. This article, as the first part of a series on pediatric CML, aims to provide precise definitions and to unify the terminology in relation to the WHO ICD codes and the methods applied for diagnosis of pediatric CML. The Ph1 chromosome and the resulting BCR-ABL1 rearrangement are the hallmarks of this leukemia and thus this entity is separated sharply from other differential diagnoses. As in adults, pediatric CML is classified by the phase of the disease and subtypes can be defined by distinct BCR-ABL1 rearrangements. The identification of the resulting different m-RNA fusion transcripts forms the basis for quantification of residual disease when monitoring the treatment response.

Pediatric CML presents at diagnosis with biological features different from adult CML pointing towards a more aggressive course of the disease. However, histopathological, cytogenetic, and immunophenotype analyses are comparable to findings in adults. At diagnosis of CML treating physicians must be alert of rarely occurring but typical clinical complications resulting from rheological problems. Finally, essential and desirable diagnostic criteria are listed which hopefully will help to unify the attempts when approaching this rare pediatric malignancy. Evidently only a clear picture of all relevant features can lay the basis for standardized treatment approaches.

## Figures and Tables

**Figure 1 cancers-13-00798-f001:**
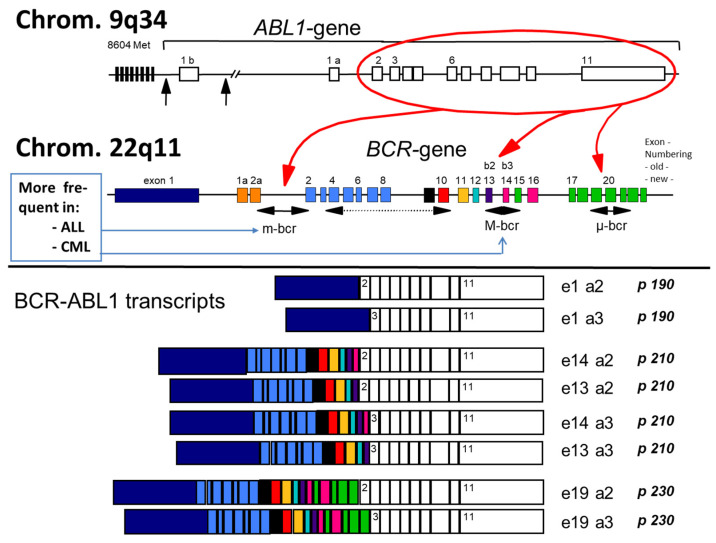
Gene breakpoints and resulting transcript types. Intronic breakpoints (vertical black arrows) of the ABL1 gene and intronic breakpoints of the BCR gene (black horizontal arrows) and the corresponding fusion proteins (Length of exons and introns not according to scale). Pseudoexons 1a and 1b on the ABL-gene as well as pseudoexons 1a and 2a on the BCR-gene are spliced out. In CML, the most frequently observed M-bcr breakpoint and the less frequently and only rarely observed breakpoints m-bcr and µ-bcr, respectively, can produce eight m-RNA fusion transcripts and translated proteins because of alternate splicing of the ABL1-gene exon 2 and because of two internal breakpoints in intron 13 and intron 14 of the BCR-gene. Additional breakpoints (indicated by a dotted horizontal arrow in this cartoon) have been described rarely or only as single case.

**Figure 2 cancers-13-00798-f002:**
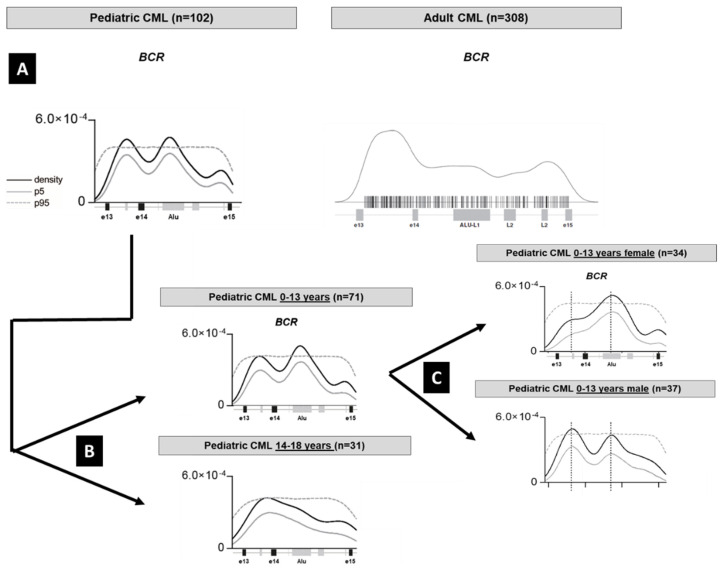
Illustration of the genomic DNA breakpoint distribution in the BCR gene and Kernel density analysis. For details see reference [59]. (**A**) When the pattern in 102 pediatric patients with CML is compared to 308 adult patients with CML as reported in the literature, pediatric CML shows a bimodal distribution which is also found in Ph1 positive acute lymphoblastic leukemia. (**B**) The bimodal “ALL-type” breakpoint can be found more frequently in prepubertal children and (**C**) in the prepubertal cohort is detectable more frequently in boys. In the Kernel plots the gray lines denote the 5% and the dotted line the 95% confidence intervals.

**Figure 3 cancers-13-00798-f003:**
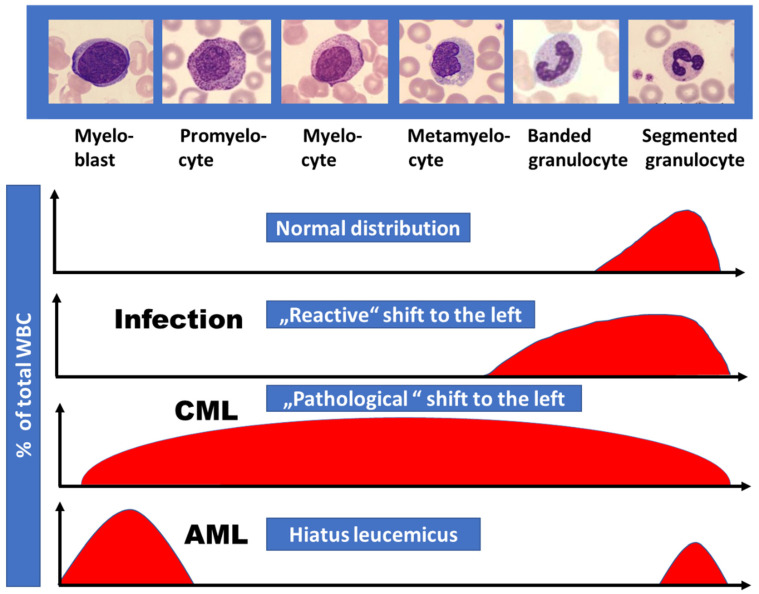
Proportion of immature granulopoietic cells (“shift to the left”) as observed in the differential white cell count in different diseases. Compared to the reactive shift to the left in bacterial infectious diseases, CML is characterized by the presence of more immature granulopoietic cells in addition including myeloblasts. In AML “hiatus leucemicus” (Latin, meaning “leukemic gap”) usually is observed exhibiting myeloblasts and mature granulocytes only.

**Figure 4 cancers-13-00798-f004:**
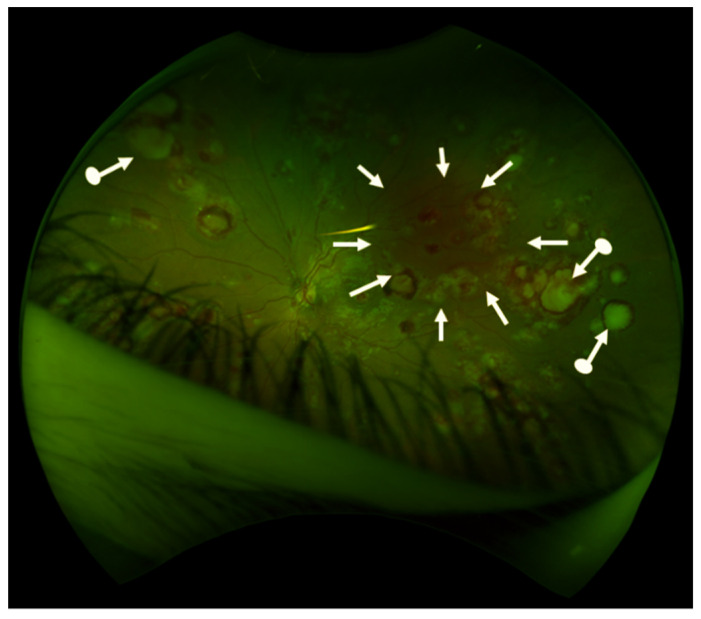
Retinal findings in a 15-year old girl who presented with blurred vision, massive splenomegaly and leukocytosis. A diagnosis of CML-AP was made. Retina is the most commonly involved intraocular structure in CML. Fundus showed retinal hemorrhages in both eyes (white arrows) and multiple white centered infiltrates “cotton-wool spots” (arrows with dot, not all infiltrates are marked). Photography (OPTOS retinal camera) by courtesy of F. Gekeler, Stuttgart).

**Figure 5 cancers-13-00798-f005:**
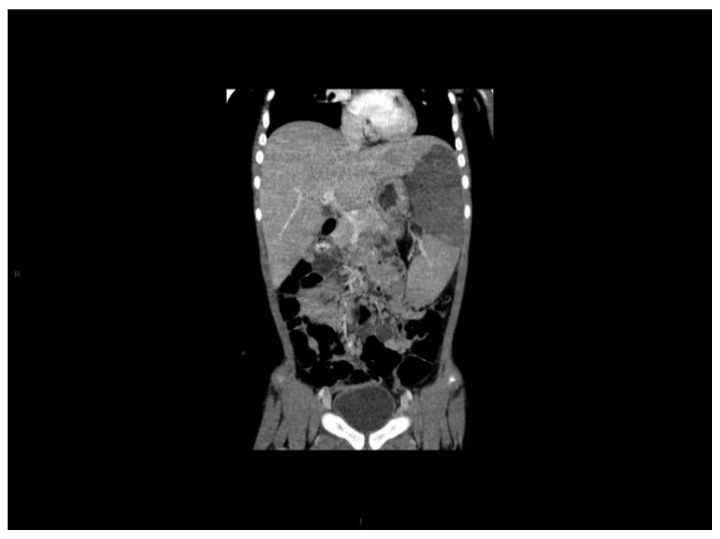
Splenic infarction at diagnosis of CML. A seven years-old girl presented with abdominal pain, splenomegaly and massive leukocytosis. Coronal CT after contrast injection showed an enlarged spleen with a large central hypodensity, well limited, with low enhancement and rectilinear borderlines. Photography by courtesy of Prof. Arnaud Y. Petit, Paris, France. For details and outcome see reference [90].

**Figure 6 cancers-13-00798-f006:**
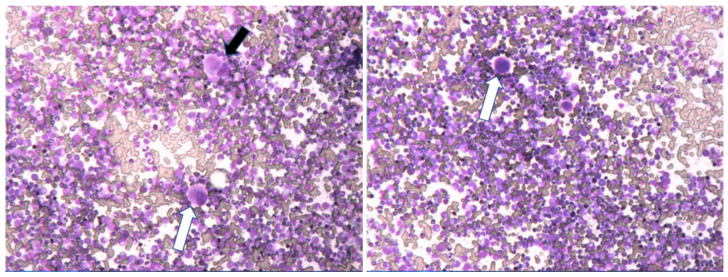
Significant hypercellular marrow at diagnosis of CML-CP showing also small hypolobulated megakaryocytes (black arrow) and micromegakaryocytes (white arrows). Magnification 80×.

**Figure 7 cancers-13-00798-f007:**
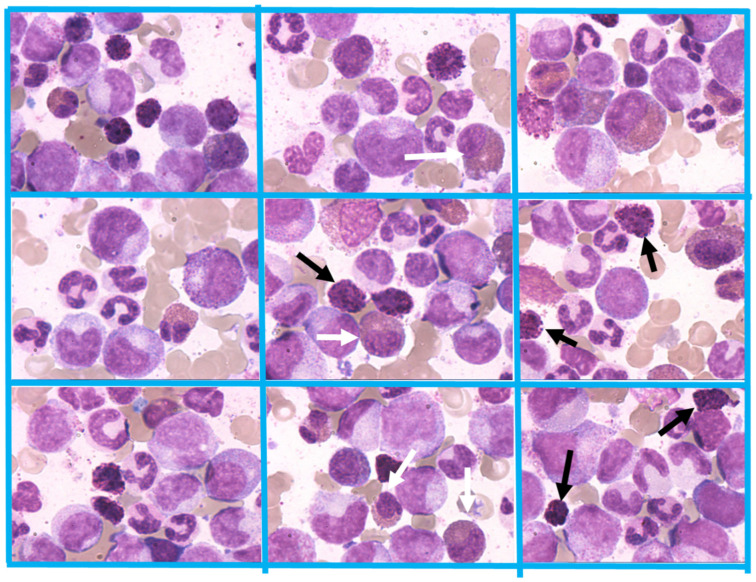
CML-CP presenting with characteristic hyperplasia of the granulocytic lineage with increased number of precursor cells (left-shifted granulopoiesis). In addition, typical immature basophils (black arrows) and immature eosinophils (white arrows) are present. Magnification 800×.

**Figure 8 cancers-13-00798-f008:**
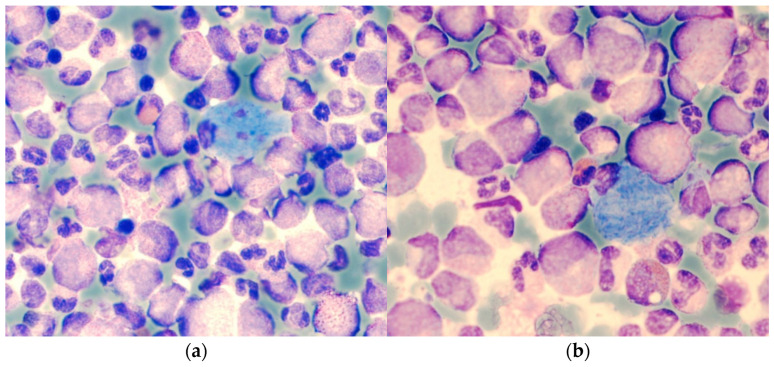
Bone marrow smears showing deep-blue Pseudo Gaucher cells. Magnification 800×.

**Figure 9 cancers-13-00798-f009:**
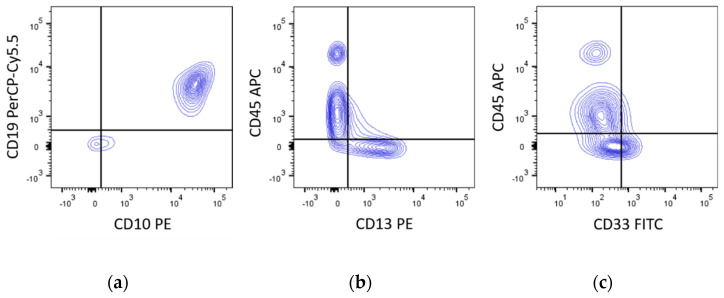
FACS analysis of CML-BP-lymphoid (CD10, CD19) with co-expression of myeloid (CD13, CD33) markers.

**Table 1 cancers-13-00798-t001:** Synonyms of chronic myeloid leukemia (CML), BCR-ABL1 positive.

Synonym	Abbreviation	Comment
CML, Philadelphia chromosome positive	CML, Ph1 chromosome+	*Nota bene*: Ph1 stands for the word “Philadelphia” only *
CML, t(9;22)(q34;q11)	not applicable	
chronic granulocytic leukemia, BCR-ABL1	CML, BCR-ABL1	The abbreviation “CGL” should be avoided
chronic granulocytic leukemia, Philadelphia chromosome positive	CML, Ph1 chromosome+	The abbreviation “CGL” should be avoided
chronic granulocytic leukemia, t(9;22)(q34;q11)	CML, t(9;22)(q34;q11)	The abbreviation “CGL” should be avoided
chronic myelogenous leukemia, BCR-ABL1 positive	CML, BCR-ABL1+	-
chronic myelogenous leukemia, Philadelphia-chromosome positive	CML, Ph1 chromosome+	-
chronic myelogenous leukemia, t(9;22)(q34;q11)	CML, t(9;22)(q34;q11)	-

* The abbreviations”, “Ph^1^”, “Ph-1”, “Ph 1” standing for “Philadelphia” can be found in the older literature and should not be used any longer.

**Table 2 cancers-13-00798-t002:** Categorization of CML by the current codes of the International Classification of Diseases (ICD-11).

ICD-11 Code	Category of CML	Comment
2B33.2	Chronic myeloid leukemia, not elsewhere classified	only to be designated in cases with incomplete diagnostics
2A20.0Y	Other specified chronic myeloid leukemia, BCR-ABL1-positive	e.g., CML, BCR-ABL1-positive, in complete remission
2A20.0Z	Chronic myeloid leukemia, BCR-ABL1-positive, unspecified	e.g., no information on the phase of CML
2A20.00	Chronic myeloid leukemia with blast crisis	-
2A20.01	Chronic myeloid leukemia, Philadelphia (Ph1) chromosome positive	-
2A20.02	Chronic myeloid leukemia, t(9:22)(q34;q11)	-

**Table 3 cancers-13-00798-t003:** Comparison of the criteria established by the ELN and the WHO for definition of the phase of CML.

Definition as Used by		
Phase	European LeukemiaNet (ELN)	World Health Organization (WHO)
CML CP	<10% blasts in PB or in BM No criteria fulfilled for CML-AP or CML-BP	<10% blasts in PB or in BM No criteria fulfilled for CML-AP or CML-BP
CML-AP	Persistent thrombocytopenia (<100 × 10 ^9^/L) unrelated to therapy >20% basophils in the PB 15–29% blasts in the PB and/or BM Sum of myeloblasts and promyelocytes >30% in the PB or BM with proportion of blasts <30%	Persistent or increasing WBC (>10 × 10^9^/L), unresponsive to therapy Persistent or increasing splenomegaly, unresponsive to therapy Persistent thrombocytosis (>1000 × 10^9^/L), unresponsive to therapy Persistent thrombocytopenia (<100 × 10^9^/L) unrelated to therapy >20% basophils in the PB 10–19% blasts in the PB and/or BM Additional clonal chromosomal abnormalities (ACA) in Ph1 cells at diagnosis that include “major route” abnormalities (second Ph1, trisomy 8, isochromosome 17q, trisomy 19), complex karyotype, or abnormalities of 3q26.2 Any new clonal chromosomal abnormality in Ph1 positive cells that occurs during therapy
CML-BP	≥30% blasts in the blood, marrow or both Extramedullary infiltrates of leukemic cells (with the exception of spleen and liver)	≥20% blasts in the blood or BM the presence of an extramedullary accumulation of blasts (with the exception of spleen and liver) (As the onset of lymphoid BP may be quite sudden, the detection of any bona fide lymphoblasts in the blood or marrow should raise con-cern for a possible impending lymphoid BP, and prompt additional laboratory and genetic studies to exclude this possibility).

**Table 4 cancers-13-00798-t004:** Age adjusted and age specific incidence rates (per 1,000,000 children) of chronic myeloproliferatice diseases (all races, males and females) depicted from the SEERS database listing data from the USA in the years 2011–2015. [45].

Age (Years)	0–14	0–19	<1	1–4	5–9	10–14	15–19
Chronic Myeloproliferative Diseases	1.4	2.1	-	0.7	1.0	2.1	4.3

## Data Availability

Not applicable for this review.

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
