# Peer review of "Definition, Epidemiology, Pathophysiology, and Essential Criteria for Diagnosis of Pediatric Chronic Myeloid Leukemia"

_cancers, 2021, doi:10.3390/cancers13040798_

Round 1

Reviewer 1 Report

The Authors are internationally recognized leading experts in the field and provided a beautiful and comprehensive review of pediatric CML. References are appropriate and updated. The figures are particularly valuable.

I have only two minor comments:
a) the prognostic impact of the most common transcript types (i.e. e13a2 and e14a2) on the kinetics of molecular response to TKI is debated in adults, I suggest discussing this topic and the data available in pediatric patients
b) it is stated that the identification of genes influencing TKI blood serum concentration and metabolism should be a "desirable" requirement for optimal monitoring of patients, however this is not discussed in the paper; I suggest explaining, or alternatively moving the sentence in the following section ("to improve the scientific understanding of the disease") as it seems to me still a matter of research

Author Response

Dear Reviewer 1

Thank you for your positive judgement on our manuscript. 

Our responses to your comments are as follows:

a) We have added information on the most frequent transcript types e13a1 and e14a2 in the section "Subtypes of CML" on page 5 and discussed the findings on prognosis and outcome when treating CML in the corresponding Section on page 16. Three additional references were cited and included in the list of references in the revised version of the manuscript.

b) We fully agree with your suggestion that the identification of genes influencing TKI blood serum concentration should be a topic of future research. Thus, the sentences has been shifted to the following section

All changes applied to the manuscript have been marked with yellow text marker for convenient identification and tracing back. 

With kind regards

Meinolf Suttorp

Reviewer 2 Report

Dr Suttorp and colleagues provide a very extensive review on the topic of pediatric CML.  They comprehensively review the diagnosis, definitions, and classification of the very rare disease of pediatric CML.

I found the article to be extremely well written, well organized, accurate, and easy to understand.  The writing is very well cited.

Dr Suttorp is a well known leader in this field, and probably the leading figure on pediatric CML in the world.

I have no major edits.

I would consider adding in treatment of pediatric CML, but I understand that this appears to be beyond the scope of this article.

Tiny type: "summery" -> "summary" in the abstract.

Author Response

Dear Reviewer 2,

Thank you positive judgement on our manuscript. We apologize for the orthographic mistake in the section Abstract and have corrected the text (summary instead of summery).

All changes applied to the text are marked in yellow to allow tracing back easily.

The coauthors and myself are planning a consecutive article with a focus on the treatment of pediatric CML. We have separated the work into two two parts in order to keep the length of each manuscript within a reasonable limit.

Kind regards

Meinolf Suttorp